# Impacts of Mesoscale Eddies on Biogeochemical Variables in the Northwest Pacific

**Jianhua Kang [†], Yu Wang [†], Shuhong Huang, Lulu Pei and Zhaohe Luo ***

Third Institute of Oceanography, Ministry of Natural Resources, Xiamen 361005, China
* Correspondence: luozhaohe@tio.org.cn
† These authors contributed equally to this work.

**Abstract:** Mesoscale eddies play an important role in regulating biogeochemical cycles. However, the response of biogeochemical variables to cold and warm eddies has not been well elucidated, mainly due to most previous studies relying on remote sensing techniques and lacking in situ observations below the surface water. Here, we used hydrographic and biochemical data from one survey in the northwestern Pacific to document the vertical biogeochemical structure of one cold and two warm eddies. We first compared the changes of key variables in the eddy core relative to eddy outside, explained the role of key layers (the mixing depth, pycnocline, nutricline, euphotic) in causing these changes, and then analyzed the main environmental factors affecting chlorophyll *a* (Chl*a*) and phytoplankton communities. Finally we focused on the response mechanisms of key biogeochemical variables to the cold and warm eddies. The results showed that biological variables (Chl*a*, microphytoplankton, picophytoplankton), salinity, dissolved inorganic nitrogen (DIN), dissolved inorganic phosphate (DIP), and dissolved inorganic silicate (DSi) in the cold eddy core increased by 0.2–134%, while in the warm eddy core, they decreased by 0.2–70% relative to the eddy outside. The cold and warm eddies were able to force the deep chlorophyll maximum (DCM), which rose or fell with the pycnocline, nutricline and euphotic depth ($Z_{eu}$) as a whole. Cold eddies with a raised thermocline could lead to about 20 m elevated DCM and enhanced phytoplankton biomass when the nutricline and thermocline were coincident. In contrast, warm eddies drove isopycnals downward, resulting in a 10–25 m drop in DCM and a decrease in nutrient and Chl*a* concentrations at the center of the eddies. The significant difference in the vertical structure of the phytoplankton community between the center and the outside of the eddy might be explained by the direct influence of both nutrient concentrations and stoichiometry changes. The contribution of microphytoplankton to total biomass was much smaller than that of picophytoplankton in oligotrophic waters where the DIN:DIP and DSi:DIN ratios are significantly low. Compared to nutrients, photosynthetically active radiation (PAR) might not be the main factor controlling phytoplankton biomass and abundance attributed to $Z_{eu}$ being consistently deeper than the mixed depth ($Z_m$), whereas it was likely to be the key limiting factor affecting the vertical distribution of the phytoplankton community.

**Keywords:** mesoscale eddy; phytoplankton; ecology; nutrient; biogeochemical; northwest Pacific

## 1. Introduction

Mesoscale eddies are physical oceanographic phenomena of water mass turbulence or circulation, which vary at a wide range of spatial (tens to hundreds of kilometers) and temporal (days to months) scales. Despite slow vertical movement ($10^{-4}$–$10^{-5}$ m/s), mesoscale eddies can significantly change the vertical distribution of nutrients in the upper ocean [1], therefore altering biological production in the euphotic zone. Mesoscale eddies play an important role in biogeochemical variables and processes [2]. Depending on the uplift or sinking of the permanent and seasonal thermocline on the vertical structure, the eddies can be classified into three types, namely cyclonic (cold) eddy, anticyclonic (warm) eddy and mode-water eddy [3].



Specifically, seawater isopycnal rise induced by cold eddies in the vertical direction may bring deep cold water rich in nutrients and $CO_2$ to the euphotic zone, causing nutrient redistribution and increasing the partial pressure of carbon dioxide ($\rho CO_2$) at the sea surface, and then releasing $CO_2$ into the atmosphere; furthermore, phytoplankton can use nutrients for photosynthetic carbon sequestration to enhance biomass and primary productivity and reduce sea surface $\rho CO_2$, while also exporting organic carbon to the deep sea and elevating deep-sea carbon burial [4]. In contrast, warm eddies are clockwise water mass circulations that increase sea level, usually with downwelling at the cores. Driven by this downward flow, the isocline is concave, and nutrients are subsequently carried away from the euphotic zone [5], resulting in a decline in phytoplankton biomass and a consequent change in community structure [6].

Mesoscale eddies have frequently been observed in the northwestern Pacific [7,8]. In the past few decades, studies on the biogeochemical effects of cold and warm eddies have gradually become the focus of research, especially regarding the effects on marine phytoplankton biomass [9–11]. Mesoscale eddies are in a constant dynamic change on spatial and temporal scales, leading to the fact that their biogeochemical impact is not only difficult to study but is also often accompanied by some conflicting views [7]. The first contradictory point concerns the response of phytoplankton Chlorophyll *a* (Chl*a*) to mesoscale eddies. Many previous studies have shown that warm eddies transport nutrients downward, resulting in lower biomass and phytoplankton, while eddy suction in cold eddies forces nutrients upward, promoting biomass and phytoplankton growth [12]. However, some researchers compared the phytoplankton Chl*a* and the community structure between two warm eddies of different origins in the northern South China Sea during the winter of 2003/2004 [13]. They found that Chl*a* was similar between two warm eddies as well as between the warm eddies and non-eddy area, although the phytoplankton community structures were different. These apparent differences from previous studies were attributed to the physical dynamics of the lateral flow and the different origins of the warm eddies.

The second paradoxical point concerns the response of the phytoplankton community to mesoscale eddies. Phytoplankton communities and their abundance have been shown to shift with eddy position [14]. Previous studies indicated that the occurrence of a higher diatom biomass in cold-core eddies during their maturation is a widely recognized phenomenon [15,16]. By comparing the observed cold eddies in the Atlantic Ocean with the results from the Bermuda time series station, the researchers found that during the period 1998–2003, cold eddies in the developmental phase frequently showed higher Chl*a*, with a higher proportion of *Prochlorococcus* (*Pro*) and a decrease in *Synechococcus* (*Syn*) [17]. Correspondingly, during the recessional phase, Chl*a* concentrations in the center of the cold eddies decreased significantly, and the edges of the cold eddies began to show higher biomass and productivity with an increase in *Syn*, while no significant elevation in biomass was found for diatoms and dinoflagellates. This difference in phytoplankton community response above is mainly caused by the different stages of development of the cold eddies.

Finally, it is also controversial to which key layers caused by the eddy would have a significant impact on the biological variables along the water column. One report pointed out that the Chl*a* concentrations in the south of the Kuroshio extension are closely related to the large-scale Rossby waves and eddy activities and are the function of the isopycnal depths [18], while another study suggested that the location of the mixed layer was decisive for the high winter and low spring Chl*a* in the Kuroshio extension of the western Pacific [19]. Researchers proposed two different hypotheses to explain the spring bloom phenomenon in the Oyashio zone of the western Pacific Ocean: one is the critical depth hypothesis, and the other is the disturbance–recovery hypothesis [20]. Their study suggested that not only the thermocline, pycnocline, nutricline, but also the sub-mesoscale upwelling affects the biological variables.

In summary, the response mechanism of phytoplankton to mesoscale eddies is both very important and complex, requiring specific analysis for different eddy characteristics

and their developmental stages or even for the whole vertical profile. To date, satellite-derived datasets have allowed researchers to investigate eddies and their biological roles at large spatial domains and high-frequency intervals [21–23]. In the northwestern Pacific, much of the current work on the surface layer of seawater was also based on data from satellite remote sensing [24], whereas much of the work below the surface was based on guesses from theoretical models developed by researchers. The role that eddies play in biogeochemical cycles is not yet well constrained, partly due to a lack of observations below the surface. Thus, we urgently need some in situ data of physical and biochemical, especially from vertical profiles within the euphotic zone, to comprehensively understand the response of organisms such as phytoplankton to eddies and to accurately identify their environmental impact factors and control mechanisms.

This study combines satellite and in situ observation to investigate biogeochemical and phytoplankton differences between one cold and two warm mesoscale eddies in the northwestern Pacific. We argue that most previous studies have relied on remote sensing and lacked in situ observations, leading to "conflicting views" on the impact eddies have on phytoplankton and nutrients. We emphasize that this paper is only a case study on different responses of biogeochemical variables to cold and warm eddies and may not be generalized to all mesoscale eddies. The highlights of the study are described in terms of the factors forcing the observed changes in the nutrient pool and in the structure of the phytoplanktonic community based on the mesoscale field of variability.

## 2. Materials and Methods

### 2.1. Sampling Dates and Stations

This study was conducted in the northwestern Pacific during the XT-3 cruise (10 May–14 June 2012) aboard the R/V Nan Feng. Three transects were visited during the cruise: Transect I (includes 8 stations from station C1 to C5) along 30 °N, covering one cold eddy (CE); Transect II (include 4 stations from station C7 to S5) along 28.5 °N, covering one warm eddy I (WE–I) under study; Transect III, which is located at 148 °E and extended from south (25 °N) to north (28.5 °N) starting from warm eddy II (WE-II), includes four stations (from station S7 to C7). A total of 15 stations were selected for sampling in order to construct a cross-sectional map to obtain a clearer picture of the location and profile structure of the eddies. Seven of these stations (black solid points) were those where the biogeochemical variables were determined, while the other 8 stations (cyan solid points) were those where only physical variables (temperature, salinity, density) were measured (Figure 1).

### 2.2. Sampling Procedures and Analytical Methods

Seawater was collected at 6–7 depths from 0 to 150 m using a 12-bottle rosette sampler attached to an SBE911 Plus conductivity–temperature–depth (CTD) system (the accuracy of conductivity and temperature is 0.0003 S/m and 0.001 °C, respectively). Photosynthetically active radiation (PAR) was measured with a radiometer (Biospherical Instruments, San Diego, CA, USA) calibrated for use in water and mounted on the CTD. The euphotic depth ($Z_{eu}$) has been traditionally assumed to be the depth of 1% of surface PAR [25].

The biogeochemical variables were measured on 3 replicated samples to control the data quality. Aliquots (450 mL) of seawater were filtered onto 25 mm Whatman GF/F filters for Chl*a* analysis. The filters were soaked in 10 mL 90% acetone solvent for 20~24 h in the dark at 0 °C. The Chl*a* extracted by 90% acetone was measured by the fluorometric method (the relative error (RE) of the replicate samples was ±10% at the level of 0.5 mg/m$^3$ for Chl*a* concentration) with a Turner Designs 10-Au-005-CE (detection limit of 0.01 mg/m$^3$) [26]. For microphytoplankton (Micro) quantitative analysis, the settlement was adopted [27]. Then, 10 or 25 mL of each sample was placed in a settling chamber for 24 h and then examined using a Zeiss inverted microscope under 100× or 200× magnification [28]. Phytoplankton were identified to the level of genus or species whenever possible [29–31].

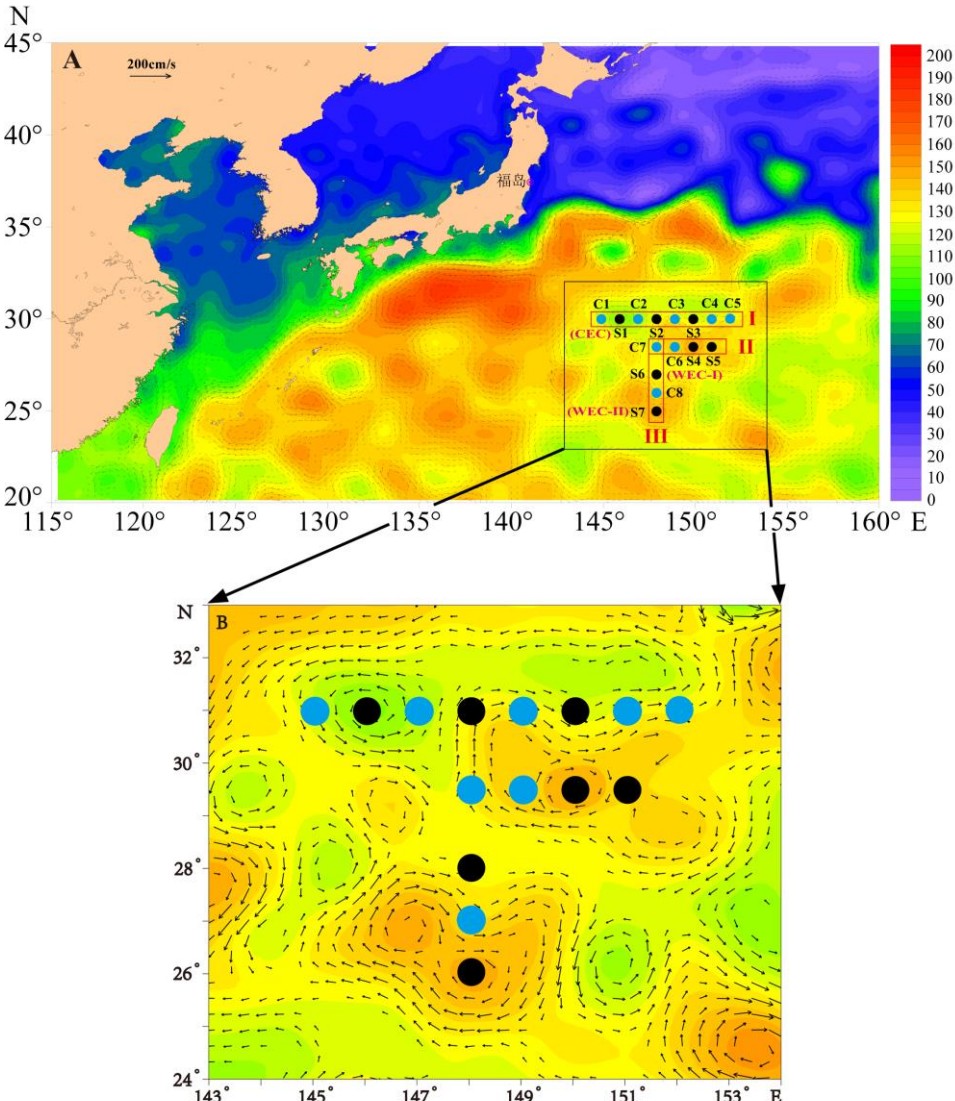

**Figure 1.** (**A**) Maps of northwestern Pacific showing the sea surface dynamic topography (cm) with geostrophic current (black vectors in cm/s). (**B**) The locations of sampling stations during the 10 to 14 June 2012 cruise are shown as solid dots, and stations for only physical (temperature and salinity, density) and biogeochemical (hydrography, biology and nutrients) variables analyses are marked in cyan and black, respectively. All stations are shown enlarged for a clearer view of their relative position to the cyclonic and anticyclonic circulation patterns.

Seawater samples for picophytoplankton (Pico) were fixed and preserved on board following the standard pretreatment methods of flow cytometry [32]. The samples were brought back to the domestic laboratory and thawed with 37 °C aqueous bath and then measured with a Flow Cytometry (BD FACS Calibur) equipped with a 488 nm laser beams (15 mW) using a pipette to take 2 mL from the thawed samples in a Falcon loading tube. The sample was measured with the speed set to 60 μL/min and the time set to 5 min. The wavelengths of fluorescence detection were green fluorescence FL1 (530 ± 21 nm), orange fluorescence (FL2, 575 ± 21 nm) and red fluorescence (FL3, >650 nm). Yellow-green fluorescent beads (Polysciences) of 1 μm size were added as an internal standard. *Pro*, *Syn* and picoeukaryotes (PEuks) were classified and quantitatively analyzed with their fluorescence signal and scattering properties in the scatterplots of FL2 vs. FL3 and SSC vs. FL3 by Cell Quest software [33].

Chemical variables including pH, dissolved oxygen (DO), suspended particulate matter (SPM), nitrate ($NO_3$-N), nitrite ($NO_2$-N), ammonium ($NH_4$-N), dissolved inorganic

nitrogen (DIN), dissolved inorganic phosphate (DIP), and dissolved inorganic silicate (DSi) were measured following the methods described in "Specifications for oceanographic survey-part 4: Survey of chemical variables in sea water" [34]. Zinc cadmium method was used for measuring the nitrate (detection limit of 0.05 µmol/L, RE was ± 7% at the level of 2.0 µmol/L), diazo-coupling method for the nitrite (detection limit of 0.02 µmol/L, RE was ± 5% at the level of 0.50 µmol/L), indigo blue spectrophotometric method for the ammonium (detection limit of 0.03 µmol/L, RE was ± 7% at the level of 1.00 µmol/L), phosphor-molybdenum blue colorimetric method for the DIP (detection limit of 0.02 µmol/L, RE was ± 10% at the level of 0.20 µmol/L), silicon molybdenum blue spectrophotometric method for the DSi (detection limit of 0.10 µmol/L, RE was ± 5% at the level of 4.5 µmol/L), electrode potentiometric method for the pH (Accuracy of 0.02, Precision of 0.01), iodometric method for the DO (detection limit of 5.3 µmol/L, the standard deviation was ± 2.8 µmol/L for DO < 160 µmol/L and ± 4.0 µmoL/l for DO ≥ 550 µmol/L), and gravimetric method for the SPM (detection limit of 4 mg/L, RE was ± 10% at the level of 5 mg/L). Paired samples t test were used to indicate the key biogeochemical variables between the cold and warm eddy with SPSS statistics Ver. 17.0. A p value less than 0.05 was significant.

The satellite altimeter data used in this paper were obtained from quasi-real-time material provided by the French Space Agency (AVISO) with a spatial resolution of 0.25° and a temporal resolution of 1 d. Altimeter raw data for merging include Jason-3, Sentinel-3A, HY-2A, Saral/AltiKa, Cryosat-2, Jason-2, Jason-1, T/P, ENVISAT, GFO, ERS1/2 (downloaded from the website: http://marine.copernicus.eu/services-portfolio/access-to-products; downloaded on 20th July 2022). The quasi-real-time data have been corrected for tides and sea surface pressures. The merged method uses the Data Unification and Altimeter Combination System (DUACS) with a 20-year mean sea level reference period (1993–2012) and Cartesian projection [35]. In this study, the sea surface dynamic topography (SSDT) was calculated within the cruise time (10 May–14 June 2012) to obtain the average and geostrophic current during the cruise (Figure 1).

Remotely sensed data (Chl*a*, sea surface temperature (SST) and sea surface salinity (SSS), PAR) were extracted from the NASA MODIS-Aqua climatology (2012) data product (https://oceandata.sci.gsfc.nasa.gov/; downloaded on 3rd August 2022). All four types of data were further processed and read using SeaDAS 8.2.0 (https://seadas.gsfc.nasa.gov/).

## 3. Results

### 3.1. The Cold and Warm Eddy Characteristics

The positions of three eddies (CE, WE–I and WE–II) during our observations were discernible from the SSDT obtained from quasi-real-time material provided by the French Space Agency (AVISO) (Figure 1A). The maximum SSDT for WE–I and WE–II during our observations was as high as 150 cm, compared to 100 cm for CE. The counterclockwise and clockwise circulations of these surface geostrophic currents confirmed their cyclonic and anticyclonic characteristics, respectively (Figure 1B). The selected CE has a diameter of about 200 km, with station S1 (146 °E, 30 °N) as the cold eddy core (CEC), station C2 as the cold eddy edge (CEE) and station S2 (148 °E, 30 °N) as the reference station outside the cold eddy (CEO); WE–I has a diameter of about 400 km, with station S4 (150 °E, 28.5 °N) as the warm eddy I core (WEC–I), station S5 (151 °E, 28.5 °N) as the warm eddy I edge (WEE–I) and station S3 (150 °E, 30 °N) as the reference station outside the warm eddy I (WEO–I); WE–II has a diameter of about 400 km, with station S7 (148 °E, 25 °N) as the warm eddy II core (WEC–II) and station S6 (148 °E, 27 °N) as the reference station outside the warm eddy II (WEO–II).

As illustrated in Figure 2, there was an obvious difference in hydrography between cold and warm eddies. In the center of the cold eddy, both isothermal and isopycnal are clearly raised upward to the shallowest depth of 20 m, which may lead to differences in water temperature and density between the center of the cold eddy and the surrounding waters (Figure 2A,B); for example, comparing the integrated values within $Z_{eu}$, the CEC (station S1) is 3.0 °C cooler and 0.8 kg/m$^3$ denser than the CEO (station S2). In contrast, the

downward displacement of the isothermal and isopycnal water in the center of the warm eddies begins to appear at the surface and ends at a maximum depth of 125 m (Figure 2C–F); for example, WEC–I (station S4) is 3.4 °C warmer and 0.6 kg /m$^3$ less dense than WEO–I (station S3); WEC–II (station S7) is 2.5 °C warmer and 0.6 kg /m$^3$ less dense than WEO–II (station S6) within $Z_{eu}$.

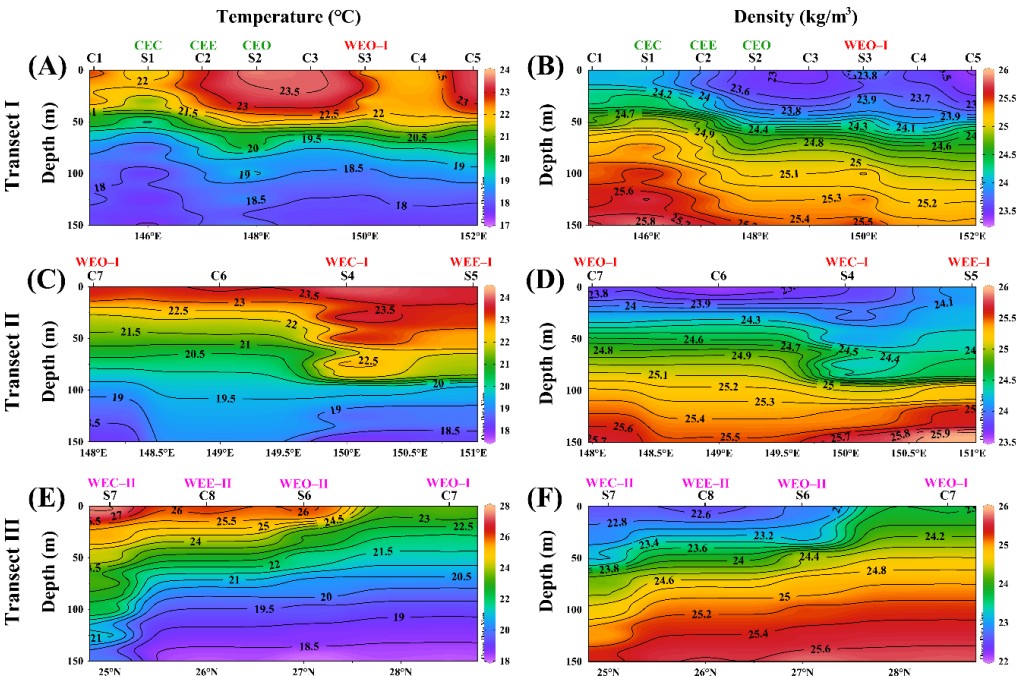

**Figure 2.** Sectional distribution of the hydrography in the upper 150 m. Distributions of (**A**) temperature, (**B**) density along transect I, (**C**) temperature, (**D**) density along transect II, (**E**) temperature, and (**F**) density along transect III. The stations are denoted at the X axis and are separated into eddy cores, edges, and outside stations at the top (CEC, CEE, WEC–I, WEE–I, WEO–I, WEC–II, WEE–II, WEO–II).

### 3.2. Biogeochemical Variables in Different Eddies

3.2.1. Comparisons of Stations

Figure 3 illustrates the integrated average and standard deviation of biogeochemical variables within the euphotic zone at different stations. The results showed that the distribution of the six chemical variables (DIN, DIP, DSi, pH, DO, SPM) was irregular; the highest and lowest values of nutrients occurred at the same position. For instance, the highest values of DIN (0.79 ± 0.06 μmol/L), DIP (0.15 ± 0.03 μmol/L) and DSi (3.04 ± 0.16 μmol/L) were found in the CEC (Station S1), and the lowest values of DIN (0.14 ± 0.02 μmol/L), DIP (0.04 ± 0.01 μmol/L) and DSi (1.48 ± 0.15 μmol/L) were found in the WEC–I or WEC–II (stations S4 or S7). The distribution pattern of biological variables (Chl*a*, Micro, Pico, *Pro*, and PEuks) was relatively uniform, with the CEC (station S1) having the highest Chl*a* (0.89 ± 0.13 mg/m$^3$), Micro (888 ± 147 cells/mL), Pico (47,333 ± 7733 cells/mL), *Pro* (39,333 ± 9105 cells/mL), PEuks (2007 ± 374 cells/mL), followed by the CEO (station S2) and WEO (station S3), WEE–I (station S5) in descending order, finally with the WEC–I (station S4) or WEC–II (station S7) being the lowest (Chl*a* (0.19 ± 0.04 mg/m$^3$), Micro (322 ± 52 cells/mL), Pico (15,177 ± 2880 cells/mL), *Pro* (11,495 ± 1691 cells/mL), PEuks (487 ± 96 cells/mL)). The results of the bar chart imply that the seawater in a cold eddy becomes warmer and warmer while salinity, density, nutrients and biomass become lower and lower as it moves from inside to outside. The contrary is true for the warm eddies.

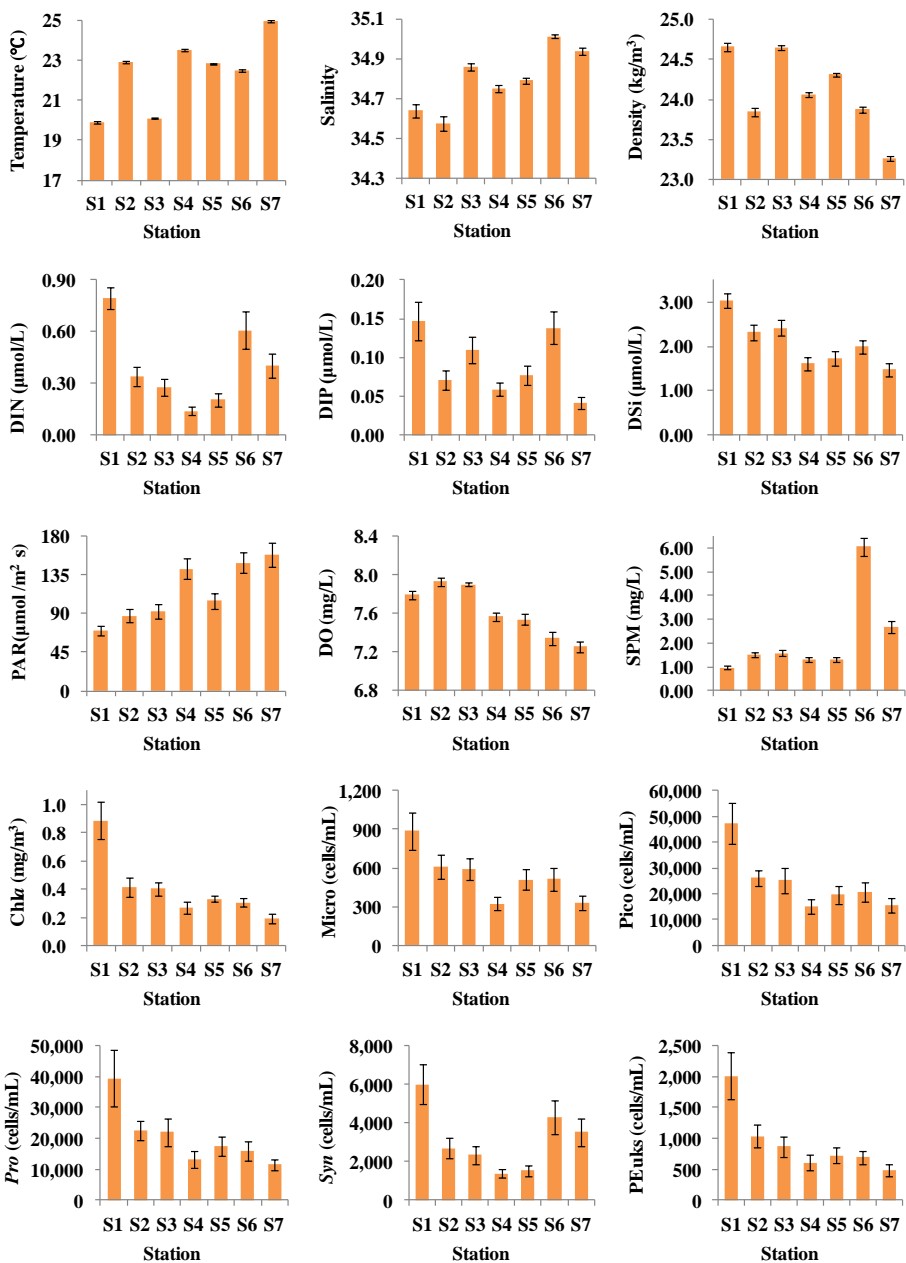

**Figure 3.** Depth-integrated mean and standard deviation of biogeochemical variables within the euphotic zone at different stations in the northwestern Pacific.

### 3.2.2. Cold Eddies

Figure 4A illustrates the percentages of increase (or decrease) in the averaged values within $Z_{eu}$ of the biogeochemical variables in the CEC relative to those in the CEO. Temperature, ammonium, DO, and SPM exhibited different ranges of decrease (0.1–35%) with the greatest in SPM and the least in salinity. Meanwhile, all biological (Chl*a*, Micro, Pico, *Pro*, *Syn*, PEuks) and other chemical variables (DIN, DIP, DSi) showed varying ranges of increase (31–134%). Among the biological variables, PEuks showed the maximum increase at 123%, while Micro presented the minimum at 46%, and among the chemical variables, DIN recorded the maximum increase (134%) and DSi with the minimum at 31%. Only one chemical parameter (pH) hardly changed between CEC and CEO. Overall, the effect of cold eddies on biogeochemical variables is all-encompassing, and dynamic interactions between the biological, geological and chemical components of the environment could often be observed in cold eddies.

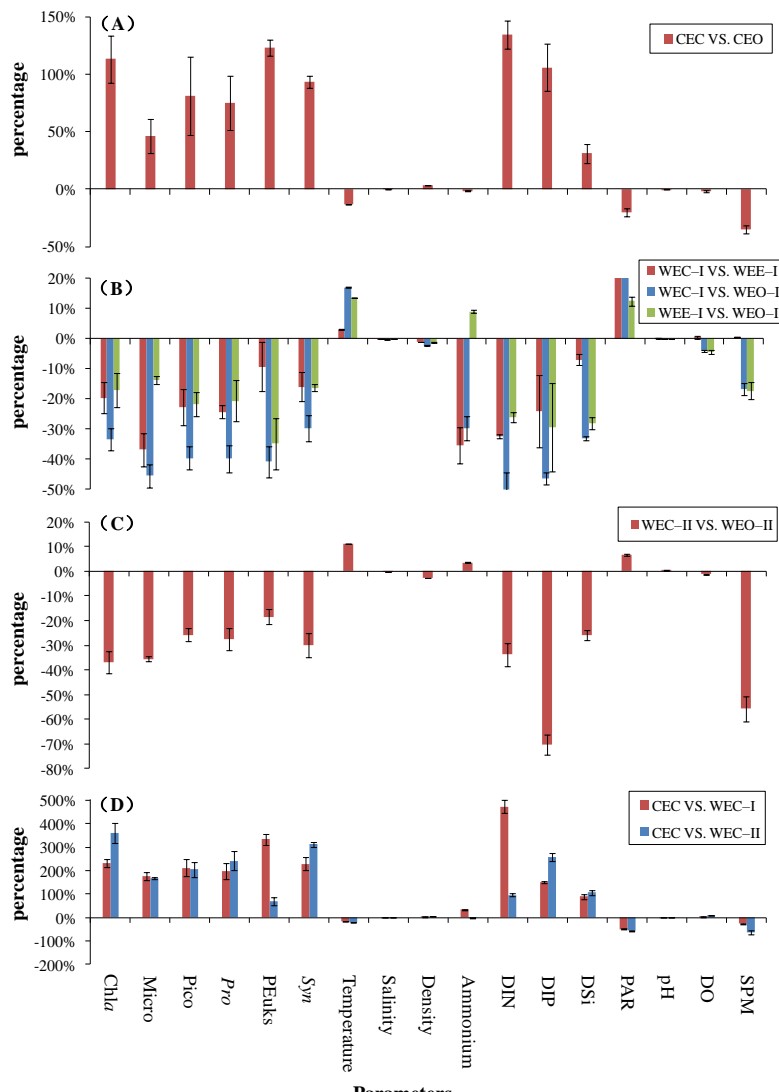

**Figure 4.** The percentages of increase (or decrease) of the averaged biogeochemical variables within the euphotic zone ((**A**). CEC vs. CEO; (**B**). WEC–I vs. WEE—-I, WEC–I vs. WEO–I, WEE–I vs. WEO–I; (**C**). WEC–II vs. WEO–II; (**D**). CEC vs. WEC–I, CEC vs. WEC–II).

### 3.2.3. Warm Eddies

Figure 4B,C illustrates the percentages of increase (decrease) in the biogeochemical variables of the WEC relative to those in the WEE or WEO. In the warm eddy from outside to the core, only the temperature and PAR showed an increasing pattern, while all other geochemical and biological variables showed a decreasing pattern, and the pH value was almost constant. The parameter with the highest percentage of decrease in the WEC–I compared to WEE–I was Micro (37%), and the lowest percentage of decrease was salinity (0.1%). The parameter with the largest gap between WEC–I and WEO–I is DIN, which decreases by up to 50%, and the smallest gap is salinity, which declines by only 0.3%. In addition, WEE–I has the highest decrease percentage (35%) and the least percentage change (0.2%) in salinity for all variables compared to the reference station in the WEO–I.

In Warm Eddy II, we compared the changes in the WEC–II variables to the WEO–II variables. The results showed that temperature, ammonium and PAR increased by 11%, 3%, 7%, respectively; meanwhile, all biological variables (Chl*a*, Micro, Pico, *Pro*, *Syn*, PEuks) and other geochemical variables (salinity, DIN, DIP, DSi, DO, SPM) showed different ranges of decrease (0.2 to 70%). Among the biological variables, Chl*a* showed the most with 37% and PEuks the least with only 18%, while among the chemical variables, DIP reached the

maximum decrease at 70%, and salinity the minimum decrease at only 0.2%. In addition, the density decreased by 3%, and the pH hardly changed CEC and CEO. Overall, the magnitude of the variation between the WEC vs. WEO is not as pronounced as that of the CEC vs. CEO.

### 3.2.4. Comparisons between Cold and Warm Eddies

Figure 4D illustrates the variation of biogeochemical variables in the CEC relative to the WEC. The results show a moderate decrease in temperature (15% and 20%), a slight rise in density (2% and 6%), and a wide range of increase in nutrients (DIN, DIP, DSi) and biological variables (Chl*a*, Micro, Pico, *Pro*, *Syn*, PEuks) of 89–473% and 71–361%, respectively, with a relatively small range of variation (increase or decrease) in other variables (salinity, ammonium, DO, and SPM) of only 0.3–64%.

Paired samples t test was used to indicate the key biogeochemical variables between the cold and warm eddies (Table 1). From the results of the comparison, no significant difference in the biological variables between WEO–I and CEO (WEO–II) was detected, which proved that S2, S3 and S6 were very suitable as reference stations. There were significant differences in some biogeochemical variables between the eddy core and outside (CEC vs. CEO, WEC–I vs. WEO–I, WEC–II vs. WEO–II), of which WEC–I shows more biological variable (Chl*a*, Pico, *Pro*, *Syn*) difference with WEO–I. The paired group with the most variables that differed was CEC vs. WEC–II, a combination that showed significant differences in all variables except for DO, SPM and *Syn*.

**Table 1.** Paired t test of biogeochemical variables between cold and warm eddies.

| Group | CEC vs. CEO | CEC vs. WEC–I | CEC vs. WEC–II | CEO vs. WEO–I | WEC–I vs. WEO–I | WEC–I vs. WEE–I | WEC–I vs. WEC–II | WEO–I vs. WEO–II | WEC–II vs. WEO–II |
|---|---|---|---|---|---|---|---|---|---|
| Temperature | **−10.815 \*** | **−6.155 \*** | **−13.485 \*** | **14.803 \*** | **6.136 \*** | **2.835 \*** | **−3.884 \*** | **−3.786 \*** | **12.536 \*** |
| | (0.000) | (0.000) | (0.000) | (0.000) | (0.001) | (0.025) | (0.008) | (0.009) | (0.000) |
| Salinity | 0.184 | −0.424 | **−8.230 \*** | **−3.344 \*** | **−4.279 \*** | **−4.661 \*** | **−8.419 \*** | **−2.716 \*** | **−5.692 \*** |
| | (0.859) | (0.685) | (0.000) | (0.016) | (0.005) | (0.002) | (0.000) | (0.035) | (0.001) |
| Density | **8.465 \*** | **4.950 \*** | **−9.070 \*** | **−6.653 \*** | **−6.844 \*** | **−10.613 \*** | **3.132 \*** | 2.047 | **−4.894 \*** |
| | (0.000) | (0.002) | (0.000) | (0.001) | (0.000) | (0.000) | (0.020) | (0.087) | (0.002) |
| DIN | **4.106 \*** | **4.200 \*** | **3.970 \*** | 1.136 | **−2.767 \*** | −0.431 | **−2.801 \*** | −0.506 | **−4.648 \*** |
| | (0.005) | (0.004) | (0.005) | (0.299) | (0.033) | (0.679) | (0.031) | (0.631) | (0.002) |
| DIP | **10.349 \*** | **7.448 \*** | **11.546 \*** | −0.467 | **−10.392 \*** | **3.976 \*** | 2.322 | **−2.483 \*** | **−11.776 \*** |
| | (0.000) | (0.000) | (0.000) | (0.657) | (0.000) | (0.005) | (0.059) | (0.048) | (0.000) |
| DSi | 0.906 | 1.814 | **4.033 \*** | 1.188 | −1.523 | −0.158 | 1.203 | **3.025 \*** | **−10.915 \*** |
| | (0.395) | (0.113) | (0.005) | (0.280) | (0.179) | (0.879) | (0.274) | (0.023) | (0.000) |
| pH | −1.895 | −2.046 | **−3.908 \*** | −0.509 | 0.375 | 1.146 | −2.293 | 0.057 | 1.927 |
| | (0.100) | (0.080) | (0.006) | (0.629) | (0.721) | (0.289) | (0.062) | (0.956) | (0.095) |
| DO | **−2.362 \*** | 1.983 | 1.730 | 0.731 | −0.513 | −1.249 | 1.302 | 1.336 | 0.158 |
| | (0.050) | (0.088) | (0.127) | (0.492) | (0.627) | (0.252) | (0.241) | (0.230) | (0.879) |
| SPM | −2.315 | −0.768 | −1.515 | −0.836 | −1.152 | −0.916 | −1.377 | −1.910 | −0.521 |
| | (0.054) | (0.468) | (0.173) | (0.435) | (0.293) | (0.390) | (0.218) | (0.105) | (0.619) |
| Chl*a* | 1.098 | 1.921 | **2.511 \*** | 0.865 | **−2.912 \*** | **−4.692 \*** | **2.953 \*** | 2.277 | −0.727 |
| | (0.309) | (0.096) | (0.014) | (0.420) | (0.027) | (0.002) | (0.025) | (0.063) | (0.491) |
| Micro | **2.469 \*** | **2.978 \*** | **3.077 \*** | −1.143 | −1.838 | −0.905 | 0.121 | 1.011 | **−3.504 \*** |
| | (0.043) | (0.021) | (0.018) | (0.297) | (0.116) | (0.396) | (0.908) | (0.351) | (0.010) |
| Pico | 1.003 | 2.096 | **2.485 \*** | 0.393 | **−3.292 \*** | **−4.923 \*** | 0.873 | 2.102 | −0.411 |
| | (0.349) | (0.074) | (0.042) | (0.708) | (0.017) | (0.002) | (0.416) | (0.080) | (0.694) |
| *Pro* | 0.759 | 1.907 | **2.404 \*** | 0.256 | **−3.111 \*** | **−4.852 \*** | 1.715 | 2.488 | −0.201 |
| | (0.473) | (0.098) | (0.047) | (0.806) | (0.021) | (0.002) | (0.137) | (0.470) | (0.847) |
| *Syn* | **2.967 \*** | **3.026 \*** | 1.755 | 1.343 | **−3.684 \*** | **−11.504 \*** | **−3.536 \*** | −3.715 | **−3.717 \*** |
| | (0.021) | (0.019) | (0.123) | (0.228) | (0.010) | (0.000) | (0.012) | (0.100) | (0.007) |
| PEuks | 1.144 | 2.034 | **2.628 \*** | 2.172 | −2.300 | **−4.361 \*** | **2.644 \*** | 1.758 | −0.435 |
| | (0.290) | (0.081) | (0.034) | (0.073) | (0.061) | (0.003) | (0.038) | (0.129) | (0.677) |

\*. Correlation is significant at the 0.05 level. The parentheses refer to the Sig. values in the paired samples t test. T values with significant differences are shown in bold.

### 3.3. Vertical Distribution in Different Eddies

3.3.1. Biogeochemical Variables

Figure 5 displays the vertical profiles of the biogeochemical variables in the eddy core in comparison with the eddy edge and eddy outside. Different characteristics of biogeochemical key layers had been presented between the CEC and CEO. Under the strong effect of the cold eddy, the mixed depth ($Z_m$), DCM, nutricline and $Z_{eu}$ in the CEC all showed significant elevation compared with the CEO. In the CEC (S1), $Z_m$ and nutricline were 30 m and $Z_{eu}$ was about 65 m, and DCM was 50 m. In the CEO (S2), $Z_m$ and nutricline were elevated to 50 m, and $Z_{eu}$ and DCM were deepened to 70 m.

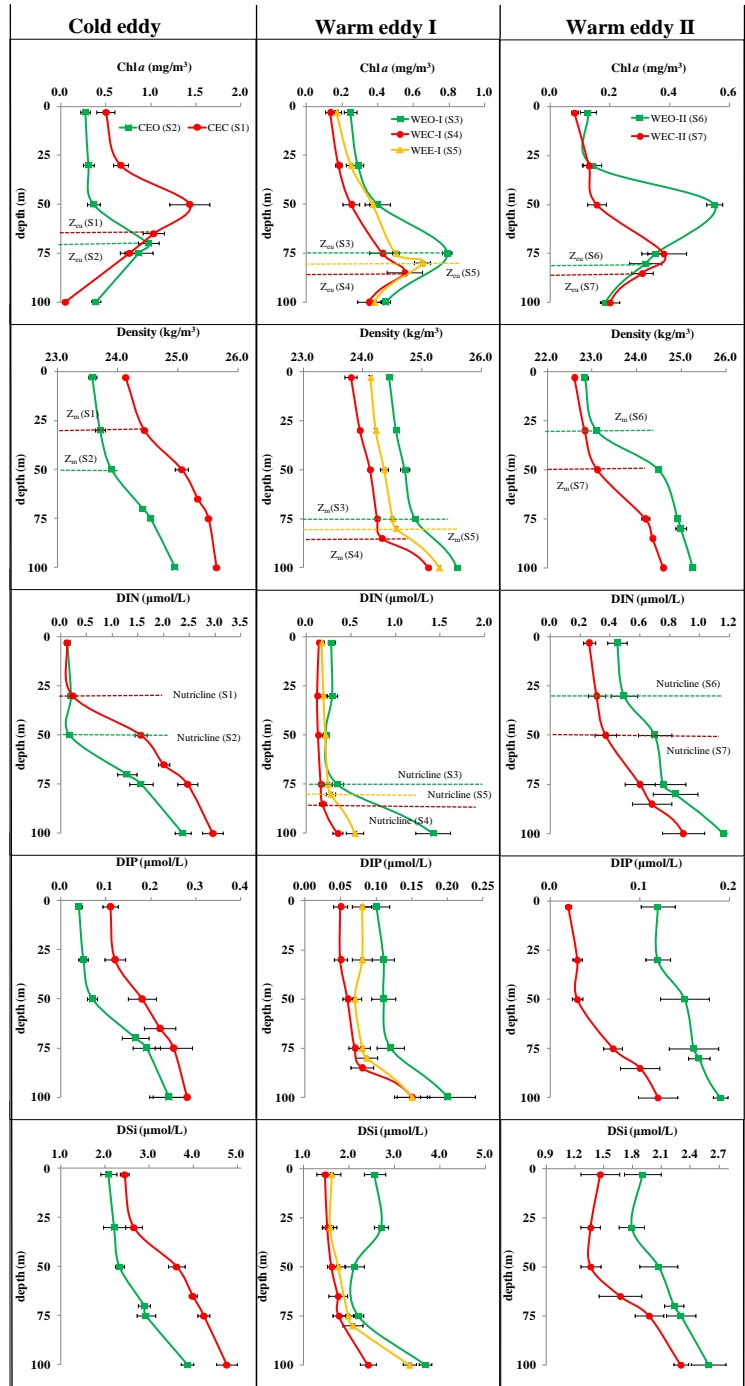

**Figure 5.** The comparison of the profiles of biogeochemical variables within the upper ocean (100 m) layer.

In contrast to the cold eddies, $Z_m$, DCM, nutricline and $Z_{eu}$ in the center of the warm eddies showed significant decreases compared to the outside of warm eddies. Each key layer showed a gradual pattern in the order of outside, edge, core. For warm eddy I, $Z_m$, nutricline, $Z_{eu}$ and DCM all showed a pattern with the depth 75–85 m in the order of outside, edge, core. In the warm eddy II, $Z_m$ and nutricline changed consistently from the outside to core, gradually decreasing from 30 to 50 m, while $Z_{eu}$ deepened from 80 to 85 m, and the DCM decreased the most, transitioning from 50 to 75 m. In conclusion, the above data results showed that cold and warm eddies were able to force the DCM, which rose or fell with the pycnocline, nutricline and $Z_{eu}$ as a whole.

### 3.3.2. Phytoplankton Community

In this research, we studied Micro and Pico, which included *Pro*, *Syn* and PEuks. The vertical profiles of phytoplankton abundance in the core and outside the mesoscale eddies are shown in Figure 6. It can be seen that there were some similarities and differences in the vertical variation of phytoplankton between the cold and warm eddies.

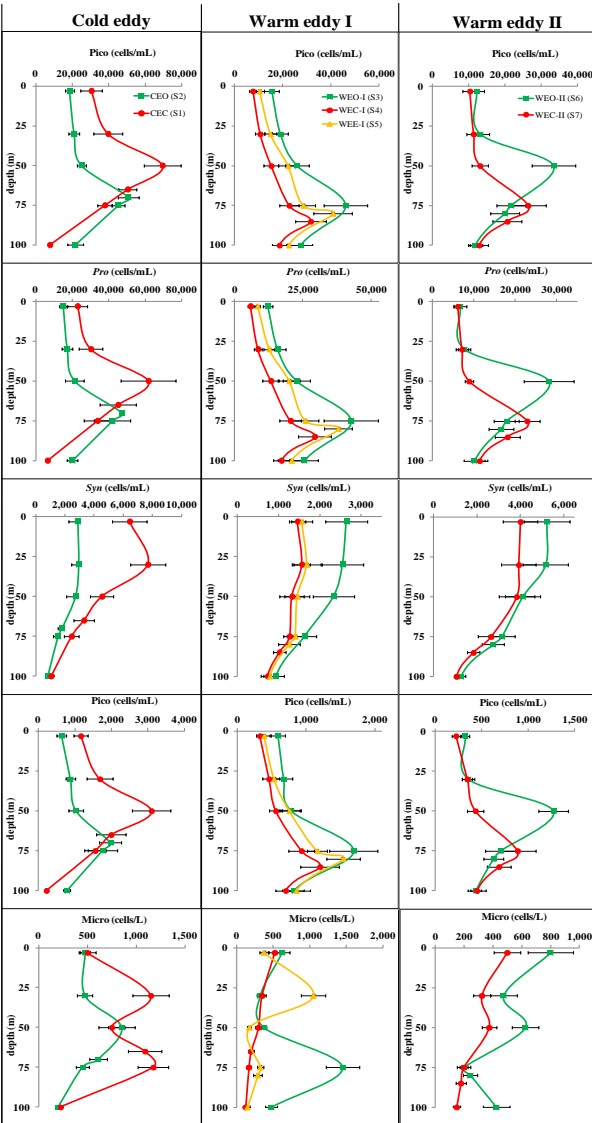

**Figure 6.** Comparison of the profiles of the phytoplankton community within the upper ocean (100 m) layer.

The first common denominator was that the order of magnitude differences among phytoplankton abundances was largely consistent. In both cold and warm eddies, average

abundances of *Pro* were generally one order of magnitude higher than *Syn* and two orders of magnitude higher than PEuks. Pico wase at least two orders of magnitude higher than Micro. Secondly, the vertical distribution of some variables basically converged. For example, the same unimodal structure was observed between Pico (including *Pro* and PEuks) and Chl*a*, which indicated that *Pro* was the dominant species of Pico while Pico was the main contributor to total biomass.

Significant differences were detected between the cold and warm eddies in terms of depth at which the maximum abundance of phytoplankton was located. For example, the maximum values of Pico, *Pro* and PEuks were all located at 75 m in the CEO (S2) and were lifted to 50 m at the CEC (S1), while for WE–I, they were located at 75 m in the WEO–I (S3), dropped by 5 m at the WEE–I (S5), continued to decline at WEC–I (S4), and finally staying at 85 m. For WE–II, they started at 50 m of WEO–II (S6), then gradually dropped from outside to inside, and ended at 75 m at the WEC–II (S7). In addition, there were large differences in the vertical distribution among *Syn*, Micro and Chl*a*. The structure of the *Syn* profile was hierarchical, with high values distributed in the surface or subsurface layers, after which the abundance gradually decreased with depth and became extremely low at the bottom of $Z_{eu}$. Micro was more complex, with different structures in different cold and warm eddies profiles. For example, it showed a bimodal structure in the CEC, with maximum values at 30 and 80 m, and a single-peaked structure in the CEO, with maximum values at 50 m; it showed a surface maximum in the WEC–I, and a single-peaked structure at WEE–I and WEO–I, with maximum values at 30 and 75 m, respectively; there was a surface maximum in the WEC–II and WEO–II, but it had a secondary peak at 50 m. The above results indicated that *Syn* and Micro, although non-major contributors to Chl*a* concentrations and different vertical structures of the profiles, still clearly showed characteristics of being influenced by mesoscale eddies. For example, the abundance of phytoplankton in CEC was significantly higher than that in the CEO, while the opposite was true for the warm eddies.

### 3.4. Relationship between Biological and Physicochemical Factors

Pearson's correlation was used to identify key environmental variables that explained the biogeographic variations in the phytoplankton community (Table 2).

**Table 2.** Pearson's correlation coefficients between environmental factors and abundance of phytoplankton (*n* = 7).

| Variables | Chl*a* | Micro | Pico | *Pro* | *Syn* | PEuks |
|---|---|---|---|---|---|---|
| Temperature | −0.777 * | −0.837 * | −0.780 * | −0.809 * | −0.397 | −0.736 |
| Salinity | −0.578 | −0.502 | −0.533 | −0.594 | −0.005 | −0.615 |
| Density | 0.672 | 0.683 | 0.625 | 0.687 | 0.090 | 0.617 |
| DIN | 0.688 | 0.686 | 0.741 | 0.661 | 0.991 ** | 0.697 |
| DIP | 0.694 | 0.766 * | 0.711 | 0.684 | 0.669 | 0.655 |
| DSi | 0.924 ** | 0.964 ** | 0.948 ** | 0.955 ** | 0.635 | 0.921 ** |
| PAR | −0.792 * | −0.859 * | −0.790 * | −0.850 * | −0.214 | −0.792 * |
| $Z_{eu}$ | −0.888 ** | −0.957 ** | −0.918 ** | −0.933 ** | −0.562 | −0.904 ** |

\* Correlation is significant at the 0.05 level (two-tailed). ** Correlation is significant at the 0.01 level (two-tailed).

In the Pearson's correlation analysis, positive correlations were observed between DIN and Syn, DIP and microphtoplantkon, DSi and biological variables (Chl*a*, Micro, Pico, *Pro*, PEuks) except for *Syn*, which indicated that most biological variables matched better with DSI, while DIN and DIP showed a poorer match with biological variables due to the high consumption of phytoplankton.

Because of the extremely low values of DIN and DIP in the study area, Pico abundance was at least two orders of magnitude higher than that of Micro, indicating that Pico is more suitable for oligotrophic waters, and its growth was not constrained by low nutrient concentrations. Furthermore, $Z_{eu}$ showed a significant negative correlation with all phytoplankton except for *Syn*, indicating that the abundance of Micro, *Pro* and PEuks decreased

with increasing $Z_{eu}$, which indirectly showed the response of biological variables to eddies, as $Z_{eu}$ was easily regulated by cold and warm eddies. Consequently, water, light and nutrient availability were important factors in explaining the influence of phytoplankton by eddies, whereas temperature and salinity explained relatively little.

### 3.5. Comparison between Satellite and In Situ Observations

The temperature, salinity, PAR and Chl*a* data acquired via satellite and in situ were compared to highlight the importance of subsurface observations of environmental variables in response to eddies (Figure 7).

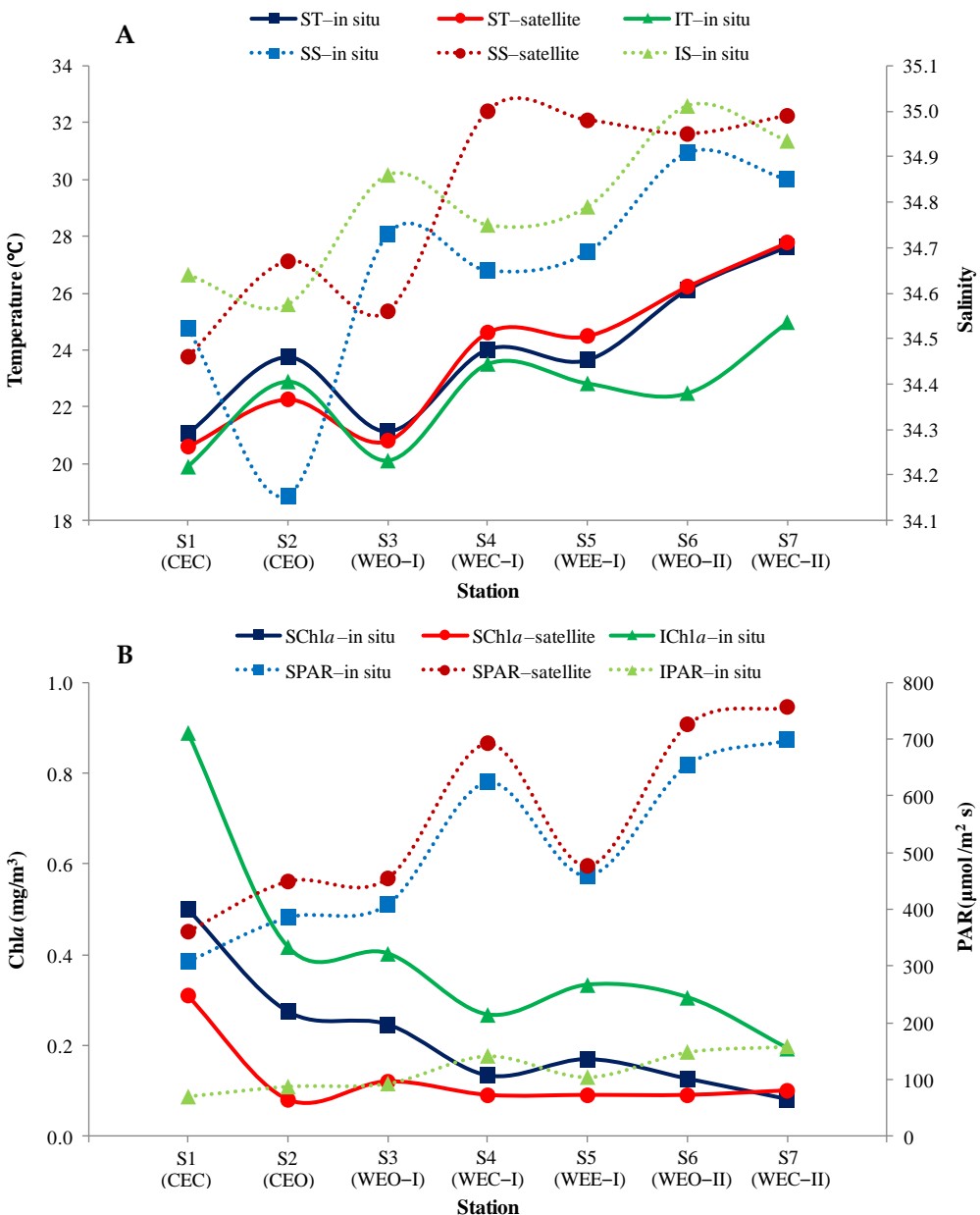

**Figure 7.** Trends in biogeochemical variables ((**A**). temperature and salinity; (**B**). Chl*a* and PAR) acquired via satellite and in situ (ST: surface temperature; SS: surface salinity; IT: integrated temperature; IS: integrated salinity; SChl*a*: surface chlorophyll *a*; IChl*a*: integrated chlorophyll *a*; SPAR: surface photosynthetically active radiation; IPAR: integrated photosynthetically active radiation).

The variation patterns of ST (SPAR) satellite, ST (SPAR) in situ and IT (IPAR) in situ were found to be identical. Meanwhile, the effects of temperature and PAR response to eddies were observed to be consistent between satellite and in situ (e.g., lower values for cold eddies and higher values for warm eddies). However, this was not the case for the SS satellite and SChl*a* satellite (Figure 7A,B). These two data obtained by satellite remote sensing were different from the in situ measurements. For example, the variation of SChl*a*-satellite was very small (0.08–0.12 mg/m$^3$) except for station S1, and they could not reflect the biological effects of the warm eddies, while both SChl*a* in situ and IChl*a* in situ were more variable (0.08–0.27 and 0.19–0.42 mg/m$^3$, respectively), and the effects of the warm eddies can be seen (Figure 7B). In addition, the trend of SS satellite is even completely opposite to SS in situ and IS in situ (Figure 7A). The analysis of the above four variables suggests that remotely sensed data may only partially match the in situ data due to algorithms or other interferences. The results also indicate that many variables, especially biological factors, attributed to their main activity in the subsurface within the euphotic zone, may not be able to accurately show their effects on eddies through surface data obtained by remote sensing methods, and instead, still need to be interpreted by profile data in situ.

## 4. Discussion

### *4.1. Response of Geochemical Variables to Cold and Warm Eddies*

Mesoscale eddies had a significant influence on the flux of nutrients into the surface or deep ocean in oligotrophic ecosystems [17]. The most important process by which mesoscale eddies influenced vertical nutrient delivery was the vertical mixing by eddy pumping [36]. Previous studies suggested that mesoscale eddies could significantly alter the vertical flux of nutrients by shifting the pycnocline [37], and about 20–40% of the nutrients demanded to support the functions of marine ecosystems were provided by the mesoscale eddies [5]. Researchers found that the nutrients in the CEC near Hawaii were twice those in the edge [38]. A previous study found that the center of warm eddies formed in the southeastern Indian Ocean could have a mixed layer (275 m) much deeper than the edge of warm eddies, but the nutrients were much lower than the edge of warm eddies [39]. Our results also showed that the nutrients in the warm eddy were lower than those in the non-eddy areas. The convergence of deep seawater in cold eddies leads to the elevation of pycnocline, and the cold and nutrient-rich water intrudes into the euphotic zone. By contrast, the sink of surface seawater in warm eddies leads to the decline of pycnocline, the warm and nutrient-depleted water penetrates further into the deep.

In this study, compared with the CEO, CEC has a 20 m elevation both in $Z_m$ and in nutricline, and a 5 m lift in $Z_{eu}$ (Figure 5), which causes a 3.0 °C lower temperature, 0.06 higher salinity and 0.8 kg m$^{-3}$ greater density in the CEC compared with CEO (Figure 3), and a 134%, 106%, and 31% increase in DIN, DIP and DSI, respectively (Figure 4A). On the contrary, for WEC–I and WEC–II compared to WEO–I and WEO–II, $Z_m$ and nutricline both deepened by 10 and 20 m, and $Z_{eu}$ decreased by 10 and 5 m, respectively (Figure 5), which caused the WEC to be 2.5–3.4 °C warmer, 0.08–0.11 lower salinity, 0.6 kg m$^{-3}$ less dense than the WEO; DIN, DIP, and DSI respectively decreased by 34–50%, 46–70%, and 26–33% (Figure 4B,C). This indicates that the eddies in this study not only alter the pycnocline, but also regulate the depth of key layers such as $Z_m$, nutricline and $Z_{eu}$, causing significant changes in all physicochemical variables in the center and edges of eddies compared to the outside of eddies.

### *4.2. Response of Biological Variables to Cold and Warm Eddies*

#### 4.2.1. Response of Chl*a* to Eddies

The displacement of DCM changed the distributions of nutrients, which in turn enhanced or decreased the phytoplankton Chl*a* [17,40]. It has been shown that the cold eddy near the Hawaiian Islands could increase the DCM Chl*a* by 2.2 times [41]. By contrast,

some researchers have reported that warm eddies in the north Atlantic deepen the density layer, resulting in changes in the phytoplankton community and a reduction in Chl*a* [14].

Our results were consistent with the general response of Chl*a* to mesoscale eddies as earlier determined. In this study, the DCM was elevated by 20 m, and Chl*a* increased by 113% in the CEC, while the DCM declined 10 and 25 m, and Chl*a* decreased 34% and 37% in the WEC–I and WEC–II, respectively, compared to the outside of eddies. The above results were mainly due to the fact that the DCM would rise and fall with the pycnocline and nutricline as a whole [17]. Previous studies have reported that the uplift (decline) in DCM and increase (decrease) in Chl*a* would make the euphotic zone shallower (deeper) [16,42]. This finding indicated the concurrent changes of nutrients, Chl*a*, and light availability in the eddy.

The biomass of the same eddy type might vary depending on the stage of development or origin of the eddy itself. For example, the cold eddies observed in the northwest subtropical Atlantic usually presented a thick DCM and high Chl*a* values during the development period, while the Chl*a* values in the center of the cold eddies decreased substantially during the recession period, followed by the onset of higher biomass and productivity in the marginal regions of cold eddies [17]. In addition, the source of eddies might also be one of the main factors affecting biomass concentration. For instance, in the northern South China Sea, warm eddies with Kuroshio or oligotrophic region origins were dominated by *Pro*, whereas warm eddies with shelf or origins within the eutrophic zone were dominated by Haptophyceae, with some degree of difference in Chl*a* concentration between these two origins [13]. In contrast to previous studies, our results suggested that the eddies were in development rather than in recession, and that they originate in oligotrophic regions where *Pro* was the main contributor to phytoplankton biomass.

In addition, although Chl*a* differed between CEC and CEO, nutrient concentrations were both very low; especially, nitrate and nitrite were not even detectable (<0.02 µmol/L). Nutrients were injected into oligotrophic waters through physical processes and then rapidly decreased as the phytoplankton population entered an exponential growth phase, the reaction time was only from one to several days [1]. The cold eddies that formed in the study area in mid-May were observed on 10 June, and the nutrients might have been depleted more than ten days before samples were collected for this study, which might explain the high biomass and low nutrient concentrations observed in the CEC.

### 4.2.2. Response of Phytoplankton to Eddies

Mesoscale eddies had a strong influence on the community structure of phytoplankton [4,43]. It was previously reported that Pico would dominate during the onset of warm eddies when nitrate supply was inadequate because Pico required relatively lower nutrient levels than Micro or even larger phytoplankton, while cyclonic cold eddies might lead to the accumulation of large phytoplankton species in Pico-dominated waters [38]. For example, diatoms could remain co-prosperous in cold eddies as well as Pico [15]. In the case of tropical and subtropical open waters, where Pro represented up to 82% of phytoplankton primary productivity, Pico could always dominate regardless of cold or warm eddies [44]. Thus, the distribution of Pico groups and their responses to physical processes (especially mesoscale eddies) had been examined in focus and had attracted increasing attention in recent years [45,46].

Our research results in the northwestern Pacific showed that among all contributors to phytoplankton biomass, the least contributor was Micro (5%), the second contributor was nanophytoplankton (17.5%), and the most contributor was Pico (77.5%). The co-prosperity of diatoms and Pico was not found in the cold eddies, but Micro (mainly diatoms and Haptophyceaes) increased or decreased slightly with the cold or warm eddies to some extent, which was basically consistent with previous studies. Thus, we still take Pico as the focus of this study. We found that the average abundance of *Pro* in the eddy core and eddy outside accounted for 74% and 75% of the total abundance of Pico, respectively, demonstrating that the eddies did not change the dominance of *Pro* (Figure 6).

However, there were significant differences in the vertical structure of the phytoplankton community between the center and outside of the mesoscale eddy. For example, the abundance of all phytoplankton in this study was significantly higher in the CEC and was lower in the WEC than in CEO and WEO, respectively. The maximum layers of *Pro* and PEuks were elevated by 25 m in the CEC and decreased by 10–25 m in the WEC. It is known that high *Syn* and PEuks abundance can be observed in areas dominated by coastal currents or upwelling [47]. However, in oligotrophic open waters without the physical input of new nutrients, nutrient availability may be important for planktonic growth. Previous studies on physical processes had shown that *Pro*, *Syn* and PEuks alternated in dominance in different water bodies or circulations because their hydrological and chemical properties, such as temperature, salinity and nutrients, might be important controls on phytoplankton distribution [48], and the combined effect of these factors ultimately led to different phytoplankton communities between the center and outside of the eddy [49]. In addition, vertical physical processes appeared to be critical to the alteration of nutricline [50], which was usually deepened by downward movement of the warm eddy or was shoaled by the upward movement of cold eddy. The vertical shift of nutricline could enhance or reduce the dominance of Pico by making eutrophic water more oligotrophic or oligotrophic water more eutrophic [51].

The vertical distribution of the Pico community and its habitat were also related. The abundance maxima of *Pro* and PEuks occurred mainly at the bottom of $Z_{eu}$ and above the nutricline in the ocean basin. *Pro* has been reported to have higher growth rates at low light irradiances, while *Syn*, however, prefers to live on the surface or in subsurface layers at relatively high light irradiances [52]. Thus, the vertical structure of the abundance of *Pro* and PEuks was easily regulated, while *Syn* was not greatly affected by mesoscales.

### 4.3. Main Physicochemical Factors Affecting Biological Variables

4.3.1. Nutrient Concentrations

As mentioned earlier, nutrient availability has a regulatory effect on the spatial pattern of phytoplankton in different marine ecosystems. As can also be seen in the vertical distribution of this study, the maximum layers of phytoplankton biomass and abundance both follow the change in depth of the nutricline (Figures 5 and 6).

It has been reported that Chl*a* and *Pro* were often positively correlated with all nutrient factors [48], but the correlation analysis in this paper showed that Chl*a* and phytoplankton (except *Syn*) were not correlated with DIN and DIP, but were only positively correlated with DSi, which might be due to the fact that DIN and DIP were easily consumed by phytoplankton in the upper oligotrophic ocean. *Syn* and PEuks were able to tolerate low temperatures and were more adapted to intermediate nutrient concentrations; thus, they were most abundant in coastal areas of the South China Sea [53], which was the reason that their abundance was at least 1–2 orders of magnitude smaller in the oceans compared to *Pro*. *Pro* was present in high abundances at low nutrient concentrations (Figures 5 and 6), consistent with adaptation to growth under low nutrient availability [54]. It is generally accepted that elevated nutrient concentrations and heavy metals caused by coastal currents or upwelling might have physiologically toxic effects on *Pro* recruitment [55]. Since these toxic effects were relatively weak in the pelagic ocean, *Pro* in oligotrophic open waters was more adapted, making its abundance usually higher as well (Figure 6).

The "Iron Hypothesis" has been systematically proven by mesoscale iron addition experiments mainly applicable to high-nutrient, low-chlorophyll (HNLC) regions of the ocean [56,57]. In situ Fe-enrichment experiments in the HNLC regions such as the subarctic Pacific have shown that the addition of dissolved iron to ocean water could significantly increase nutrient utilization of phytoplankton and has led to an enhancement in the concentration of chlorophyll [58,59]. However, recent studies indicate that in the NPSG, iron is not the most limiting factor affecting the growth of micro phytoplankton. In the NPSG, the primary replenishment mechanism of nutrients in surface waters is through the diffusion and vertical mixing of deep nutrient-rich waters and/or lateral advection [60]. In contrast,

the input of iron is mainly via atmospheric dust deposition [61,62], while the inputs of iron from below the euphotic zone are negligible. Therefore, in the NPSG, the effect on nutrient alteration is considered significantly higher than that of iron.

### 4.3.2. Nutrient Stoichiometry

To assess the status of nutrient limitation at each station, the concentration criteria (DIN < 1.0 µmol/L, DIP < 0.1 µmol/L) proposed by Fisher et al. [63] and ratio criteria (1. if DIN: DIP < 10 and DSi: DIN > 1, DIN limitation; 2. if DSi: DIP > 22 and DIN: DIP > 22, DIP limitation; 3. if DSi: DIP < 10 and DSi: DIN < 1, DSi limitation) proposed by Justić et al. [64] were used. The assessment results indicated varying degrees of DIN limits in the studied waters. In the study area, not only do all stations show DIN limitation, but 57% of the stations also showed DIP limitation (Table 3). The DIN and DIP co-limitation might indicate extreme nutrient depletion in the sea area due to insufficient nutrient supplementation following phytoplankton overconsumption. The DSi was a better match for the Chl*a* pattern. Due to the relative abundance of DSi within the euphotic zone, phytoplankton were not DSi limited, and their pattern well visualized the effects of the eddy on the vertical structure of nutrients and Chl*a* (Figure 5).

**Table 3.** Nutrient stoichiometry and limitation at all stations in the northwestern Pacific.

| Eddy | Station | DIN | DIP | DSi | DIN:DIP | DSi:DIP | DSi:DIN | Limitation |
|------|---------|-----|-----|-----|---------|---------|---------|------------|
| CEC | S1 | 0.79 | 0.15 | 3.04 | 5.40 | 20.70 | 3.84 | DIN |
| CEO | S2 | 0.34 | 0.07 | 2.32 | 4.75 | 32.57 | 6.86 | DIN and DIP |
| WEO–I | S3 | 0.28 | 0.11 | 2.41 | 2.52 | 21.96 | 8.72 | DIN |
| WEC–I | S4 | 0.14 | 0.06 | 1.61 | 2.38 | 27.38 | 11.51 | DIN and DIP |
| WEE–I | S5 | 0.20 | 0.08 | 1.73 | 2.65 | 22.44 | 8.47 | DIN and DIP |
| WEO–II | S6 | 0.61 | 0.14 | 1.99 | 4.41 | 14.44 | 3.28 | DIN |
| WEC–II | S7 | 0.40 | 0.04 | 1.48 | 9.78 | 35.95 | 3.67 | DIN and DIP |

In addition to nutrient concentrations, the nutrient stoichiometry of eddies seems to play an important role in determining the structure of the phytoplankton community. We observed distinct nutrient limitation between the core and outside of eddies (DIN limitation occurred in the CEC and WEO, DIN and DIP co-limitation happened in the WEC and CEO), which suggested that different phytoplankton communities had been growing. DIN:DIP was lower than or equal to the Redfield ratio (16:1), indicating that at least part of the phytoplankton community was composed of diatoms [65,66]. However, excessively high or low DIN:DIP might cause a shift from large to small phytoplankton [67]. It was reported that, under phosphate-depletion conditions in the South China Sea, an average DIN:DIP molar ratio of 42 could shift the dominant species in the phytoplankton community from diatoms to dinoflagellates and cyanobacteria [45]. The contribution of Micro to total biomass was much smaller than that of Pico in oligotrophic areas where the DIN:DIP ratio was too low [61]. Thus, the low DIN:DIP ratio caused by the low DIN concentration in this study may have influenced the composition of the phytoplankton community between the core and outside of the eddy, such as the decrease in diatom and the increase in Pico.

In addition, DSi:DIN in upwelling or downwelling waters could be important in determining the structure of phytoplankton communities sustained by mesoscale eddies [68], which might also affect the abundance of Micro. The DSi:DIN was very different from the standard Redfield ratio (1:1), both indicating a significant reduction in DSi [66]. In this study, DSi:DIN ratios calculated for CEC and WEC–II were about 3.84 and 3.67, respectively, and this low DIN:DIP means that Micro, such as the diatom, is almost impossible

to dominate in these oligotrophic waters. Previous studies illustrated that changes in phytoplankton size structure within mesoscale eddies might be influenced by variations in nutrient concentrations and stoichiometry [69]. Therefore, it is reasonable to infer that the community structure dominated by Pico rather than by Micro observed under mesoscale eddies is a reflection of the combined effect of low nutrient supply and stoichiometry.

### 4.3.3. Photosynthetically Active Radiation

A negative correlation between the abundance of most biological variables (Chl*a*, Micro, Pico, *Pro*, and PEuks) and PAR ($Z_{eu}$) were observed (Table 2), which contradicts our previous common knowledge that higher PAR or deeper $Z_{eu}$ contributes to greater phytoplankton biomass and abundance [21]. In this study, $Z_{eu}$ was always deeper than the mixed layer in both cold and warm eddies (Figure 5), which ensured that the nutricline was always within or close to the bottom of $Z_{eu}$, thus ensuring that phytoplankton could be unrestricted by light in the nutricline. Meanwhile, satellite and in situ data show a clear opposite trend between surface PAR and Chl*a* during the station transition from northern to southern latitudes (Figure 7). Therefore, even though light was important, it was not a decisive factor in the horizontal distribution controlling phytoplankton biomass and abundance within $Z_{eu}$.

However, PAR was a key limiting factor affecting the vertical distribution of phytoplankton communities in oligotrophic waters [70]. It has been proven that *Pro* was very sensitive to UV light, and its low fluorescence populations were dominant in the upper layers of the water column, while high fluorescence populations still occurred at 150 m or deeper depth [59,71]. This is not the case with *Syn*, which is more adapted to high light and can barely grow below the euphotic layer. PEuks are more suitable for photosynthesis in blue light than prokaryotes; thus, they can be abundantly distributed at the bottom of $Z_{eu}$ in oligotrophic waters. The typical vertical distribution pattern of nutrient-poor seas has been summarized by researchers [72]. For example, *Syn* peaked between 0 and 50 m and then decreased downward, while *Pro* peaked at 75 m and then declined slowly with depth, and PEuks gradually elevated in abundance from 75 m, peaked at 100 m, and decreased sharply thereafter. The results of the present study share many similarities with previous results [32], indicating that the adaptation to light is selective and stable for each phytoplankton group. Furthermore, nutrients were almost depleted within the euphotic zone due to the pronounced stratification in the subtropical oligotrophic gyres. Nutrient availability was sufficient below $Z_{eu}$. However, phytoplankton abundance was not greater under conditions with high nutrients in the low light region, which might be reasonably driven by light limitation (Figure 6). Thus, PAR availability might help explain the role of nutrients in regulating vertical variation in Pico.

## 5. Conclusions

Based on the in situ data measured in the cusp of spring and summer of 2012 in the northwestern Pacific, this study analyzed and elaborated the responses of biogeochemical variables under the influences of mesoscale eddies. The main conclusions are as follows:

(1) Temperature, ammonium, DO, PAR and SPM decreased by 1–35%, while most of the biological variables (Chl*a*, Micro, Pico, *Pro*, *Syn*, PEuks) and geochemical variables (salinity, density, DIN, DIP, DSi) increased by 0.2–134% in the CEC compared to the CEO. In contrast, WEC–I and WEC–II showed an increasing percentage of 7–53% in temperature and PAR and a decrease in percentage of 0.2–70% in biological variables (Chl*a*, Micro, Pico, *Pro*, *Syn*, PEuks) and other geochemical variables (salinity, DIN, DIP, DSi, DO, SPM) compared to WEO–I and WEO–II. The magnitude of variation between WEC and WEO were not as pronounced as that between CEC and CEO.

(2) The cold and warm eddies were able to force the DCM to rise or drop with pycnocline, nutricline and $Z_{eu}$ as a whole. In particular, cold eddies with a raised thermocline could lead to the DCM uplift by 20 m and enhance phytoplankton biomass when the nutricline and thermocline were coincident. Furthermore, warm eddies drove

isopycnals downward, resulting in a 10–25 m drop in DCM and a decrease in nutrient and Chl*a* concentrations at the WEC.

(3) In terms of the depth of the maximum abundance of Micro and Pico, there were significant differences between the center and outside of the eddy, but in terms of the overall vertical structure, both in the cold and warm eddies, *Pro* and PEuks maintained a unimodal pattern basically consistent with Chl*a*, *Syn* showed a surface or subsurface maximum type, while multiple types of Micro existed. The alteration of nutricline was usually deepened by downward movement of the warm eddy or was shoaled by upward movement of the cold eddy. The vertical shift of nutricline could enhance or reduce the dominance of Pico by making eutrophic water more oligotrophic or oligotrophic water more eutrophic.

(4) The significant difference in the vertical structure of the phytoplankton community between the center and outside of the eddy might be explained by the direct influence of both nutrient concentrations and stoichiometry. The contribution of Micro to total biomass was much smaller than that of Pico in oligotrophic regions where the DIN:DIP and DSi:DIN ratios were exceedingly low. PAR was not the main factor controlling phytoplankton biomass and abundance in the eddies attributed to $Z_{eu}$ being consistently deeper than $Z_m$, but it might be a key limiting factor affecting the vertical distribution of the phytoplankton community.

**Author Contributions:** Funding acquisition, Investigation, Methodology, Project administration, Writing—original draft, J.K.; Conceptualization, Writing—review and editing, Y.W.; Data curation, Formal analysis, S.H.; Data curation, Visualization, L.P.; Funding acquisition, Supervision, Validation, Writing—original draft, Z.L. All authors have read and agreed to the published version of the manuscript.

**Funding:** This work was funded by the Natural Science Foundation of Fujian, China (2019J05150, 2022J06029), the National Natural Science Foundation of China (no. 41506136), the National Key Research and Development Program of China (no. 2019YFE0124700) and the State Oceanic Administration of China Special Fund Project (no. SOA201303).

**Institutional Review Board Statement:** Not applicable.

**Informed Consent Statement:** Not applicable.

**Data Availability Statement:** The raw data supporting the conclusions of this article will be made available by the authors, without undue reservation.

**Acknowledgments:** Thanks to the members of the South China Sea Fisheries Research Institute, Chinese Academy of Fishery Sciences, and to the crews of "Nan Feng" scientific investigation ship. We sincerely thank Hala F Mohamed for her help in the English editing of the manuscripts. We also would like to express our great appreciation to the editor and to the anonymous reviewers for their constructive comments on the manuscript.

**Conflicts of Interest:** The authors declare that they have no conflict of interest.

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
