# Peer review of "Impacts of Mesoscale Eddies on Biogeochemical Variables in the Northwest Pacific"

_jmse, doi:10.3390/jmse10101451_

Round 1

Reviewer 1 Report

This manuscript (MS) is devoted to a comparative analysis of two types of mesoscale (one cold and two warm) eddies based on real (in situ) hydrographic and biochemical observational data in the northwestern Pacific. The paper presents the results of the analysis of the key variables in the core of cold and warm eddies compared to the outer part of the eddies, the roles of the key layers in the occurrence and dynamics of the processes. Overall, it can be noted that the manuscript deals with an interesting topic; the analysis is clear and the interpretation of the results is correct. I see several points that I recommend to pay attention.

1)      The use of the term “parameters” is not very good. I recommend using biogeochemical state variables. This is due to the fact that usually parameters take values from a fixed set, and there are no such restrictions for variables.

2)      Line -389 - Pearson's rank correlation coefficients between environmental and factors abundances of phytoplankton. Here you need to remove “rank”. In fact, here it is better to use Spearman's rank correlation coefficients, since they are more suitable for characterizing the relationship between variables, and will weakly depend on the accuracy of the estimation of environmental variables and the abundance of phytoplankton.

3)      Although such an important variable as salinity is presented in the MS, the changes in this variable are not reflected in the Abstract and in the Conclusion. This variable is very important for many organisms from different taxonomic groups in oceanic and marine environments.

4)      I recommend paying attention to the uniform use of abbreviations. For example, the abbreviation CEC is introduced for cold eddy core, CEO for cold eddy outside, WEC for warm eddy core, etc. After the first introduction of these abbreviations, they must be used everywhere in the future.

Author Response

Dear Reviewer,

Thank you very much for your comments, which have been crucial in improving the content and quality of our manuscript. We have taken all your key concerns very seriously and have responded to each of them to the best of our ability.

Responses to specific suggestions and comments

(1) The use of the term “parameters” is not very good. I recommend using biogeochemical state variables. This is due to the fact that usually parameters take values from a fixed set, and there are no such restrictions for variables.

A:Thank you very much for your suggestion. We have replaced “parameters” with “variables” in the full manuscript.

(2) Line -389 - Pearson's rank correlation coefficients between environmental and factors abundances of phytoplankton. Here you need to remove “rank”. In fact, here it is better to use Spearman's rank correlation coefficients, since they are more suitable for characterizing the relationship between variables, and will weakly depend on the accuracy of the estimation of environmental variables and the abundance of phytoplankton.

A:Thank you very much for your suggestion, “Rank” has been removed. In additon, the relationship between the environmental variable PAR and phytoplankton abundance has been added in the Table 2. All data were tested to be normally distributed as continuous values and can be considered as perfectly suitable for the Pearson correlation analysis. As the Spearman correlation analysis is more applicable to non-normally distributed data, we decided to retain the Pearson correlation analysis where the results are likely to be more accurate and objective.

(3) Although such an important variable as salinity is presented in the MS, the changes in this variable are not reflected in the Abstract and in the Conclusion. This variable is very important for many organisms from different taxonomic groups in oceanic and marine environments.

A:Horizontally, the variability of salinity in the ocean basin is much less significant than that of temperature and density, and this seems to be borne out by the integral averages within euphotic zone in this manuscript (e.g. range of variability between stations in the Fig. 4: salinity -0.8% to 0.2%, temperature -20% to 17%, density -3% to 6%). In this study, our main emphasis is on the response of the variable to eddy, rather than its effect on biology. It is well known that seawater density is a function of temperature, salinity and pressure, so our original manuscript focused only on temperature and density while deliberately ignoring salinity in order to streamline the text and improve readability.

No doubt your valuable comments to us are also very important, and as you say, salinity is an important variable regardless of its variability. Under your suggestion, we have added salinity in various sections throughout the manuscript: for example, the variation of salinity and its response to eddies are described in the results (Line 313, Section 3.2.1; Line 345-446, Section 3.2.2-3.2.4), the increased/decreased percentages of salinity and their error bars are added in Figure 4; The patterns of salinity variation are summarised in the Abstract (P17-19), Results (P666-679, Section 3.5), Discussion (P701-706, Section 4.1; ) and Conclusion (P974-980).

(4) I recommend paying attention to the uniform use of abbreviations. For example, the abbreviation CEC is introduced for cold eddy core, CEO for cold eddy outside, WEC for warm eddy core, etc. After the first introduction of these abbreviations, they must be used everywhere in the future.

A:Thank you very much for your valuable comments. We have been corrected the abbreviation errors in the full manuscript.

Reviewer 2 Report

Review of jmse-1889625 titled “Impacts of mesoscale eddies on biogeochemical parameters in the northwest Pacific” authored by Jianhua Kang , Yu Wang , Shuhong Huang , Lulu Pei and Zhaohe Luo.

Lou and co-authors present work is timely and relevant for global climate projection models, ocean-based carbon dioxide remediation proposals, and phytoplankton community dynamics.

The authors discuss the nuances of commonly held rules of nutrient cycling and chlorophyll a distribution in cold and warm eddies that require further investigation into physiochemical and biological parameters. The group expands the current knowledge base of mesoscale eddies nutrient cycling by incorporating subsurface observations which allows them to connect and elucidate biological and geophysical processes at higher resolution than is possible with satellite data alone. While the authors are careful not to generalize the characteristics of 1 cold and 2 warm eddies in the Northwestern Pacific to all mesoscale eddies, their measurements align with expected features of cold and warm eddies and therefore the additional associations found with phytoplankton structure and nutrient stoichiometry for example, may also be common. I recommend this article for publication with revisions.

Major revisions:

While the data holds valuable information I wonder if the authors could comment on why only pairwise T-tests were used instead of multivariate analysis which, seems more appropriate given the multiple factors examined and would pull out the most influential factors.

The authors could demonstrate a strong case for in situ observations by contrasting the results obtained with and without subsurface observations, using only satellite for example.

Minor revisions:

Regardless of the statistics used, the changes in pH are expected to be small so whether or not they are biologically relevant biologically relevant, graphs of pH and other small changes should have a second axis with a higher resolution scale.

I suggest moving table 1 to supplemental information and incorporating appropriate statistics into figures 3 and 4 and adding error bars.

The manuscript is easily understood but should be reviewed for minor errors, primarily grammar. I will propose the following correction to a statement occurs three times in the manuscript:

“The cold and warm eddies were able to force the deep chlorophyll maximum (DCM) which, rose or fell with the pycnocline, nutricline and euphotic depth (Zeu) as a whole.”

Line 306 “improvement” assumes a hierarchy of natural states which is relative.

Author Response

Dear Reviewer,

Thank you very much for your valuable comments, which have been crucial in improving the content and quality of our manuscript. We have taken all your key concerns very seriously and have responded to each of them to the best of our ability.

Responses to specific suggestions and comments

 Major revisions:

(1) While the data holds valuable information I wonder if the authors could comment on why only pairwise T-tests were used instead of multivariate analysis which, seems more appropriate given the multiple factors examined and would pull out the most influential factors.

A:Thank you very much for your comments. We changed the earlier independent sample analysis to a paired analysis for the following several reasons: Firstly, most of sampling layers at each station were consistent and suitable for paired analysis. Secongdly, Paired T-tests are a form of blocking, and have greater power  than unpaired tests when the paired units are similar with respect to "noise factors" that are independent of membership in the two groups being compared. Paired-Samples T-tests can be used to reduce the effects of confounding factors in an observational study, and this method is well suited to small samples. In contrast, the primary purpose of multivariate analysis is to find the significant effect of multiple independent variables on a dependent variable and is more applicable to larger samples. An important aim of Table 1 is to determine whether there are significant differences in the mean biogeochemical variables within and outside the same eddy or between different eddies, rather than to identify the most important factors influencing eddies. Therefore, a Paired T-test may be more applicable than a multivariate analysis in this study (redescribed in Line 447-456, Section 3.2.4).

(2) The authors could demonstrate a strong case for in situ observations by contrasting the results obtained with and without subsurface observations, using only satellite for example.

A:Your valuable suggestions have been a great help in improving the quality of the manuscript. We downloaded surface data for some of the MODISA parameters (e.g. temperature, salinity, PAR and Chla, which are relevant to this study) from the NASA website and subsequently plotted these data against the field measured values (Fig. 7). The results show that the surface values of some variables (e.g. salinity and chla) do not match their integrated averages, which not only reflect the bias of the remotely sensed data, but also confirms that the most important processes of biological activity are mainly concentrated in the subsurface within the euphotic zone, thus reinforcing the need for profiling data to analyse the eddy effect. Methods for acquiring satellite remote sensing data have been added to Section 2.2 (Line 248-251) and related content that reviewer concerns has been described in a separate Section 3.5 (Line 636-679).

  1. Minor revisions:

(1) Regardless of the statistics used, the changes in pH are expected to be small so whether or not they are biologically relevant biologically relevant, graphs of pH and other small changes should have a second axis with a higher resolution scale.

A:As you commented, the variation in pH is very small and does not correlate well with biological factors, so we replaced it with PAR in Fig. 3 and kept it in Fig. 4 for comparison with other variables. With the addition of changes in PAR (Figs. 3, 4) and its correlation with biological parameters (Table 2, Fig. 7), more evidence was obtained for the idea that the effect of PAR on biological variables differed between the horizontal and vertical dimensions (Section 4.3.3, Line 929-968).

(2) I suggest moving table 1 to supplemental information and incorporating appropriate statistics into figures 3 and 4 and adding error bars.

A:Thank you very much for your suggestion. The error bars have been added in Fig.3 and Fig.4.

(3) The manuscript is easily understood but should be reviewed for minor errors, primarily grammar. I will propose the following correction to a statement occurs three times in the manuscript:“The cold and warm eddies were able to force the deep chlorophyll maximum (DCM) which, rose or fell with the pycnocline, nutricline and euphotic depth (Zeu) as a whole.”

A:I appreciate your comments. We have invited experts to revise the full text for some minor grammatical errors.

(4) Line 306 “improvement” assumes a hierarchy of natural states which is relative.

A:Thank you very much for for your valuable suggestion, “improvement” has been replaced with “increase”(Line 443, Section 3.2.4).
